

# Climate System Response to Stratospheric Sulfate Aerosols: Sensitivity to Altitude of Aerosol Layer

Krishnamohan Krishna-Pillai Sukumara-Pillai[1], Govindasamy Bala[1], and Long Cao[2], Lei Duan[2,3] and Ken Caldeira[3]

[1] Centre for Atmospheric and Oceanic Sciences, Indian Institute of Science, Bangalore 560012, India

[2] Department of Earth Sciences, Zhejiang University, Hangzhou, Zhejiang 310027, China

[3] Department of Global Ecology, Carnegie Institution for Science, Stanford, CA 94305, USA

*Correspondence to*: Krishnamohan Krishna-Pillai Sukumara-Pillai (krishmet@gmail.com)

**Abstract.** Reduction of surface temperatures of the planet by injecting sulfate aerosols in the stratosphere has been suggested as an option to reduce the amount of human-induced climate warming. Several previous studies have shown that for a specified amount of injection, aerosols injected at a higher altitude in the stratosphere would produce more cooling because aerosol sedimentation would take longer time. In this study, we isolate and assess the sensitivity to the altitude of the aerosol layer of stratospheric aerosol radiative forcing and the resulting climate change. We study this by prescribing a specified amount of sulfate aerosols, of a size typical of what is produced by volcanoes, distributed uniformly at different levels in the stratosphere. We find that stratospheric sulfate aerosols are more effective in cooling climate when they reside higher in the stratosphere. We explain this sensitivity in terms of effective radiative forcing: volcanic aerosols heat the stratospheric layers where they reside, altering stratospheric water vapor content, tropospheric stability and clouds, and consequently the effective radiative forcing. We show that the magnitude of the effective radiative forcing is larger when aerosols are prescribed at higher altitudes and the differences in radiative forcing due to fast adjustment processes can account for a substantial part of the dependence of amount of cooling on aerosol altitude. These altitude effects would be additional to dependences on aerosol microphysics, transport, and sedimentation, which are outside the scope of this study. The cooling effectiveness of stratospheric sulfate aerosols likely increases with altitude of the aerosol layer both because aerosols higher in the stratosphere have larger effective radiative forcing and because they have a longer stratospheric residence time; these two effects are likely to be of comparable importance.

## 1 Introduction

Anthropogenic emissions of greenhouse gases alter the radiative balance of the planet, leading to long-term climate changes (IPCC, 2013). Of particular interest is the warming from an increasing concentration of atmospheric carbon dioxide ($CO_2$), which is the primary warming agent in the industrial era. Solar radiation management (also known as solar geoengineering)



through albedo enhancement methods such as deliberate injection of sulfate aerosols into the stratosphere has been suggested as an option to counteract human-induced climate change (Budyko, 1977; Crutzen, 2006). In some such envisioned geoengineering implementations, the positive radiative forcing from the greenhouse gases would be partially or fully offset by negative radiative forcing from increased shortwave scattering by aerosols.

Major volcanic eruptions have been considered as a natural analogue to stratospheric sulfate aerosol geoengineering. Major volcanic eruptions inject the sulfate aerosol precursor $SO_2$ into the stratosphere where it is converted to sulfate aerosols. These sulfate aerosol concentrations decay with an e-folding time of approximately 1 year (Robock, 2000). The increased scattering of shortwave radiation by the aerosols has a cooling effect on the climate system (Hansen et al., 1992; Robock, 2000; Soden et al., 2002). Volcanic eruptions are episodic, but for stratospheric aerosol geoengineering, the aerosol layer

would need to be maintained with quasi-continuous injection of additional aerosols. The quasi-continuous injection can lead to particle growth where the newly injected particles coagulate with background particles, which can potentially lead to decreased scattering efficiency (Heckendorn et al., 2009; Niemeier et al., 2011; Niemeier and Timmreck, 2015; Tilmes et al., 2017).

The time-evolution of radiative forcing and surface cooling per unit mass of injection of aerosols depends on several

factors such as type of aerosol used (Pope et al., 2012; Weisenstein et al., 2015), particle size (Rasch et al., 2008; Heckendorn et al., 2009), amount of aerosols injected (Heckendorn et al., 2009; Niemeier and Timmreck, 2015; Kleinschmitt et al., 2018), and the geographical location and altitude of injection (Tilmes et al., 2017; Dai et al., 2018). One of the primary factors affecting the amount of cooling in geoengineering scenarios is aerosol particle size. For a specified mass, smaller particles are more efficient in scattering (Rasch et al., 2008; Heckendorn et al., 2009). As particles with radius in the range of 0.1 µm have

the largest backscattering cross section per unit mass, they have been suggested to be the most suitable for geoengineering (Heckendorn et al., 2009).

The amount of injection, evolution of the size of the particles, and removal processes influence the radiative forcing and resulting climate change in state-of-the-art climate models which simulate the evolution and transport of sulfate aerosols. The rate of injection, location, and altitude of injection control processes such as particle growth (by nucleation, condensation,

coagulation or evaporation), transport, gravitational settling and removal processes (Heckendorn et al., 2009; Niemeier and Timmreck, 2015; Tilmes et al., 2017; Kleinschmitt et al., 2018) and strongly influence the efficiency of the stratospheric geoengineering schemes. The amount of cooling produced by continuous sulfate aerosol injection initially increases as the rate of injection increases but then decreases as the rate increases further (Niemeier et al., 2011; Niemeier and Timmreck, 2015; Kleinschmitt et al., 2018). This is because as the rate of injection increases beyond a threshold, coagulation increases, forming

larger particles. Larger particles are less efficient in scattering sunlight and are more susceptible to removal through sedimentation (Tilmes et al., 2017; Kleinschmitt et al., 2018).





The altitude of injection affects the microphysics and transport of the aerosols in the stratosphere and thereby affects the amount of cooling produced. High-altitude injection of the aerosols extends the sedimentation time and contributes to a longer aerosol lifetime in the stratosphere (Heckendorn et al., 2009: Niemeier et al., 2011; Niemeier and Timmreck, 2015). In the stratosphere, circulation patterns associated with the Brewer-Dobson Circulation are important as they influence aerosol transport and burden. Tilmes et al. (2017) show that, for equatorial regions, high-altitude injections increase total aerosol burden more than low-altitude injections because of longer sedimentation paths in the stratosphere associated with the deep branch of the Brewer-Dobson Circulation. However, the longer lifetime also leads to particles with larger effective radii that reduce the scattering effect and that sediment faster from the stratosphere. Kleinschmitt et al. (2018) find that for tropical injections the net radiative forcing is nearly independent of the altitude of injection despite an increase in the sedimentation time with the altitude of injection, due to the counteracting effects of the particle growth (and hence shorter lifetime) and the resulting reduced scattering properties.

The climates generated by stratospheric sulfate injections can be modulated by varying the timing, latitude and altitude of aerosol injection. A set of studies using the Whole Atmosphere Community Climate Model (CESM1-WACCM) showed that multiple simultaneous climate objectives could be met by strategically injecting aerosols at multiple locations in the stratosphere (Tilmes et al., 2017; Kravitz et al., 2017; MacMartin et al., 2017; Tilmes et al., 2018a). These studies have shown that the spatial pattern of aerosol optical depth (AOD) can be partially controlled by optimizing the locations of injection (MacMartin et al., 2017; Kravitz et al., 2017). Non-equatorial high-altitude injections are more efficient in controlling the surface climate because the transport of aerosols into middle and high latitudes results in particles of a smaller effective radius and larger aerosol burden in these latitudes (Tilmes et al., 2017; MacMartin et al., 2017).

The studies discussed above have studied the climate responses mainly focused on the changes in aerosol burden with particle microphysics, transport, and removal processes. But there are several other fast-adjustment processes in the climate system which can impact the effective radiative forcing and climate responses. Aerosols prescribed in the stratosphere can cause local warming in the stratosphere by absorbing near-IR and terrestrial radiation (Stenchikov et al., 1998; Ferraro et al., 2011). This warming can lead to changes in the amount of water vapor in the stratosphere (Dessler et al., 2013) and the amount of high clouds by changing the tropospheric stability (Visioni et al., 2018). Boucher et al. (2017) has shown that these fast adjustment processes can influence the effective radiative forcing of the climate system for sulfate aerosol injections. The altitude of the prescribed aerosol layer can thus affect climate depending on the proximity of the heated layer to the tropopause. Although most of citied studies include the fast adjustment processes, radiative forcing and response, there is a lack of clear and systematic understanding of the dependence of radiative forcing and climate response on the altitude of sulfate aerosols in the stratosphere.

In this study, we use idealized climate model experiments to systematically study the sensitivity of the effective radiative forcing and the simulated surface climate to the height at which aerosols are prescribed in the stratosphere. In all our



stratospheric aerosol experiments, we use the same total amount of aerosols but alter their altitude. Our idealized simulations are intended to highlight the radiative influences of aerosol height and isolate these effects from effects associated with aerosol particle evolution and transport.

## 2 Methods

### 2.1 Model details

To study the dependence of the surface climate on the height of the sulfate aerosols in the stratosphere, we use the NCAR Community Earth System Model version 1.0.4 (CESM; Gent et al., 2011). The CESM consists of five components: atmosphere (Community Atmosphere Model version 4 - CAM4), sea-ice (Community Ice Code version 4 - CICE4), land (Community Land Model version 4 - CLM4), ocean (Parallel Ocean Program - POP), and land-ice (Community Ice Sheet Model), which

are coupled through a coupler. For this study, we use a configuration where CAM4 is coupled with the Community Land Model and a Slab Ocean Model (SOM) for simulating the climate change. We also use the prescribed sea surface temperature (pSST) configuration of CAM4 for estimating the radiative forcings. The configuration used here has a horizontal resolution of 1.9° in latitude and 2.5° in longitude with 26 layers in the vertical. The top of the atmosphere (TOA) in the model is approximately at 3 hPa. The land model used here (CLM4) has an integrated representation of water, carbon and nitrogen

cycles (Oleson et al., 2010).

### 2.2 Experimental design

The reference climate of our study is based on a 100-year pre-industrial control simulation (called "1XCO2" hereafter) with the atmospheric $CO_2$ concentration fixed at 284.7 ppm. We also perform a '2XCO2' experiment where the atmospheric $CO_2$ concentration is doubled to 569.4 ppm. To assess the sensitivity to the altitude of prescribed aerosols, a set of three stratospheric

aerosol experiments are designed by altering the altitude of additional aerosol layer but keeping the total mass of aerosols constant at 20 Mt and $CO_2$ concentration at 569.4 ppm. The mass of aerosol was chosen based on Nalam et al. (2017), where they prescribed 20 Mt of background sulfate aerosols in five layers centered at 37 hPa to offset the global mean surface temperature change caused by a doubling of $CO_2$. In CAM4, the sulfate aerosols are log-normally distributed with fixed size distributions (Neale et al., 2010). For our stratospheric aerosol experiments, we use volcanic aerosols which have an effective

mean radius of 0.426 μm and a geometric standard deviation of 1.25. The mass of the aerosols consists of 75% $H_2SO_4$ and 25% $H_2O$ (Neale et al., 2010). The zonal variations as well as interannual variations (for this study) in mixing ratio of the volcanic aerosols are omitted (Ammann et al., 2003; Neale et al., 2010). The volcanic aerosol size used here corresponds to the large aerosols formed 6 to 12 months after a volcanic eruption (Stenchikov et al., 1998; Bauman et al., 2003; Rasch et al., 2008). The aerosols are distributed in single model layers centered at pressure levels 100 hPa, 70 hPa, and 37 hPa altitudes

with layer thicknesses in the range of 15.5±1.0 hPa in each case. Corresponding standard atmospheric heights are approximately 16 km, 19 km, and 22 km. These experiments are referred to as Volc_100hPa, Volc_70hPa, and Volc_37hPa.



In CAM4, the solar radiation is divided into 19 discrete spectral and pseudo-spectral intervals in the radiation module (Briegleb, 1992; Collins, 1998; Neale et al., 2010). The near IR absorption by volcanic aerosols is calculated in the model along with the longwave absorption which is a function of the seven longwave bands specified in the model. The concentration distributions of all other types of aerosols in the model such as dust, organic carbon, black carbon, and sea-salt are unchanged

in the 2XCO2 and in the three stratospheric aerosol experiments. All the slab ocean model simulations are performed for 100 years. Climate change analysis is performed on the last 60 years of model-generated data (from year 41 to 100), as the simulated climate closely approaches equilibrium within the first 20-30 years. The corresponding prescribed SST model configuration is used to simulate 60 years and the last 30 years of data are used for estimating radiative forcing and related fast adjustments.

## 3 Results

Before discussing the main results, it is instructive to briefly review the concepts of effective radiative forcing, fast adjustments, efficacy of different forcing agents, and the efficiency of aerosols. These concepts are discussed briefly in the supplementary Sect. S1 where the various methods of estimating the effective radiative forcing are also discussed: the prescribed-SST method (Hansen et al., 2005; Bala et al., 2010), the regression method (Gregory et al., 2004; Gregory and Webb, 2008), and the two-point method (Modak et al., 2018; Duan et al., 2018). Results on effective radiative forcing are presented in Sect. 3.1, and

results for the climate feedback parameter and efficacy (supplementary Sect. S1) are presented in Sect. 3.2. Throughout the Sect., the uncertainties in the global mean values of the variables of any simulation are represented by one standard deviation to show the internal variability and the uncertainties for the changes are represented by standard error.

### 3.1 Global Mean Temperature and Net Top-of-Atmosphere Fluxes

Because our sulfate simulations produce cooling from a 2XCO2 background state, in the interest of consistency we report all

results relative to the 2XCO2 simulation. Results for global mean temperature change and top-of-atmosphere (TOA) fluxes are shown in Fig. 1 and Table 1. A halving of $CO_2$ concentrations from the 2XCO2 state in the prescribed SST configuration causes a top-of-atmosphere net radiative flux ($\Delta N_{SST}$) of -3.52±0.09 W m$^{-2}$ at TOA (Fig. 1a), as found in previous studies that used the CAM4 model (Nalam et al., 2017). Due to land surface cooling, the global mean surface temperature change ($\Delta T_{SST}$) is -0.24±0.01 K in this case. Quasi-steady-state results for halving of $CO_2$ concentrations from the 2XCO2 state in the slab-

ocean-model configuration show global mean temperature change ($\Delta T_{SOM}$) of -3.13±0.03 K and TOA flux change ($\Delta N_{SOM}$) of -0.01±0.12 W m$^{-2}$ in this case.

There is more negative TOA radiative imbalance when the volcanic aerosols are prescribed at a higher altitude (Fig. 1a): $\Delta N_{SST}$ is -2.79±0.11, -3.44±0.09, and -3.91±0.11 W m$^{-2}$, for the Volc_100hPa, Volc_70hPa, and Volc_37hPa simulations, respectively, relative to the 2XCO2 control case. The corresponding global mean surface temperature changes ($\Delta T_{SST}$) in these

prescribed SST simulations are -0.13±0.01, -0.13±0.01, and -0.14±0.01, respectively. Residual TOA net fluxes in the steady-state for the sulfate aerosol slab ocean simulations ($\Delta N_{SOM}$) are 0.02±0.13, 0.01±0.13, and 0.0±0.12, for the Volc_100hPa,



Volc_70hPa, and Volc_37hPa simulations respectively. The corresponding $\Delta T_{SOM}$ values are -2.18±0.03, -2.57±0.03, and -2.91±0.03 K, respectively.

## 3.2 Radiative forcing and climate feedback parameters

With the values presented above, using Eq. (1) and (2) in Supplementary Sect. S1, we can calculate the effective radiative-
forcing (F) and climate feedback parameters (λ) for each of our experimental simulations (Fig. 1 and Table 1) using the two-
point method (Supplementary Sect. S1). A halving of $CO_2$ concentration from 2XCO2 to 1XCO2 results in an estimate for F
of -3.82±0.09 W m$^{-2}$, and an estimate for λ of 1.22±0.05 W m$^{-2}$ K$^{-1}$. Introduction of stratospheric aerosol layers at 100 hPa, 70
hPa, and 37 hPa results in estimates for F of -2.97±0.11 W m$^{-2}$, -3.62±0.09 W m$^{-2}$, and -4.12±0.11 W m$^{-2}$, respectively.
Corresponding estimates of λ are 1.37±0.09, 1.41±0.07, and 1.42±0.06 W m$^{-2}$ K$^{-1}$. Thus, there is a substantial increase in the
magnitude of radiative forcing from sulfate aerosols when they are higher in the stratosphere (Fig. 1b); this effect appears to
be slightly offset by a small (i.e., not statistically significant) increase in the climate feedback parameter. For example, for the
37 hPa case relative to the 100 hPa case, the radiative forcing is 38% larger in magnitude, but the climate feedback parameter
is 3% larger, resulting in a temperature change that is 34% larger in magnitude. The climate feedback parameters for sulfate
aerosols differs substantially from the climate feedback parameter for $CO_2$, resulting in efficacy values ($e_{SAI}$) of 0.89±0.07,
0.87±0.05, and 0.86±0.05 for the Volc_100hPa, Volc_70hPa, and Volc_37hPa cases, respectively, indicating that effective
radiative forcing from stratospheric sulfate aerosols would generate 11 to 14% less global mean temperature change than would
an equivalent amount of effective radiative forcing from $CO_2$. The magnitude of climate feedback parameter differs slightly
between the stratospheric sulfate experiments, which is mainly associated with the changes in the cloudy-sky feedback
parameters (Fig. S1). The reasons for these changes are not analyzed here but would be investigated in detail in a future study.
The larger magnitude of the climate feedback parameter obtained for sulfate aerosols relative the $CO_2$ forcing is qualitatively
similar to the difference between the feedback parameters for solar irradiance and $CO_2$ forcing found in a recent study (Modak
et al., 2016). Our calculated efficacy values for stratospheric sulfate aerosols are somewhat larger than the value of 0.83 for
solar irradiance estimated by Modak et al. (2016). This is likely due to differing climate sensitivity of the version of the
atmospheric model used (CAM5 in Modak et al. (2016) and CAM4 for our experiments) and the differing heating structures
in the stratosphere for changes in solar irradiance versus sulfate aerosols. The efficacy value estimated for sulfates in our study
is broadly consistent with that reported by Duan et al. (2018) where it is found that efficacy of sulfate aerosols at the top of the
atmosphere relative to $CO_2$ is 0.85.

We have also applied the two-point method (Supplementary Sect. S1) to the individual radiative forcing components.
The radiative forcing and corresponding feedback parameters are shown in Fig. S1 and Table S1 which indicates that the
longwave (LW) forcing from volcanic aerosols are not negligible (Fig. 2b) - the magnitude is about 13% of the SW forcing.
The total LW radiative forcing is positive in the stratospheric aerosol experiments relative to the 2XCO2 case (i.e., increased
downward LW radiation; Fig. 2b) but the negative shortwave (SW) forcing dominates (i.e., increased upward SW radiation;



Fig. 2a) and hence the net TOA radiative forcing is negative relative (increased upward) to the 2XCO2 case (Fig. 1b). A detailed analysis of the radiative forcing components and fast adjustments in clouds, water vapor and temperature are given below.

The clear-sky SW radiative forcing is negative in all cases (Fig. 2c) due to the SW back-scattering by the prescribed aerosols. The sensitivity to the altitude of aerosols can be explained by the changes in water vapor content in the stratosphere. When aerosols are prescribed at lower levels close to the tropopause, radiative heating by aerosols leads to an increase in cold point tropopause temperature and an increase in stratospheric water vapor (Fig. S2). The increase in water vapor leads to increased absorption of SW radiation, which can provide a strong positive water vapor feedback. As the changes in water vapor amount decreases rapidly in the stratosphere with height of the prescribed aerosols, the water vapor feedback related SW absorption decreases and hence we find a larger negative SW clear-sky radiative forcing when aerosols are prescribed at higher altitudes (Fig. 2c).

The clear-sky LW forcing is positive for all cases (Fig. 2d) due to LW absorption by volcanic aerosols and their differences are associated with different changes in water vapor. As discussed earlier, there is an increase in water vapor (Fig. S2) which is larger when aerosols are prescribed at lower levels. As water vapor absorbs LW radiation, we find that the LW clear-sky forcing increases when aerosols are prescribed at lower stratospheric levels.

The SW cloud radiative forcing is positive in all cases (Fig. 2e) because of a reduction in clouds in the stratospheric aerosol experiments relative to the 2XCO2 case (Fig. S3a). The upper troposphere warms in the stratospheric aerosol experiments (Fig. S4 and S5) because of mixing between the tropospheric and radiatively heated stratospheric air. This can cause a reduction in the high cloud because of the cloud burn-off effect (Ackerman et al., 2000). Also, as the upper troposphere warms, the stability of the troposphere increases and reduces the water vapor transport to the upper troposphere reducing the probability of ice-supersaturation (Visioni et al., 2018). Both these effects lead to a reduction of high clouds. Further, the increase in stability and less water vapor transport to the upper troposphere leads to an increase in low cloud for the Volc_100hPa case relative to the Vol_70hPa and Volc_35hPa cases (Fig. S3c). The increase in tropospheric stability is less when aerosols are prescribed at higher stratospheric levels as the upper tropospheric warming decreases. As low clouds are optically thicker than high clouds and their increase is larger for the Volc_100hPa case (Fig. S3c), a corresponding less positive SW cloud radiative forcing is simulated in the Volc_100hPa case (Fig. 2c).

A sensitivity of cloudy-sky LW forcing to the height of the aerosols is also simulated (Fig. 2f) which can be attributed to the changes in high cloud cover in the stratospheric aerosol experiments (Fig. S3b and S6). The decrease in high clouds results in a cirrus cloud thinning effect (Storelvmo et al., 2013) and allows more longwave radiation to pass through the atmosphere resulting in negative longwave forcing. The magnitude of this effect decreases as the aerosols are prescribed at



higher altitudes (Fig. 2f). In the Vol_100hPa case, the magnitude of the cloud-induced LW forcing is about one third of the net SW forcing which shows that the indirect high cloud effect is large.

## 3.3 Global climate change

The radiative forcing of -3.82 W m$^{-2}$ from a halving of $CO_2$ in the 1XCO2 experiment leads to a decrease in global mean surface temperature by 3.13 K (Figs. 3a and 4a). In the stratospheric aerosol experiments, the aerosol induced negative radiative forcing induces a surface cooling (Fig. 3a). As the TOA radiative forcing varies with the altitude of the aerosols in the stratospheric aerosol experiments, corresponding changes are simulated in surface temperature. The global cooling is more when volcanic aerosols are prescribed at the higher levels of the stratosphere (Fig. 3a). The spatial changes in global mean surface temperature for the stratospheric aerosol experiments relative to the 2XCO2 experiment are shown in Fig. 4. In all stratospheric sulfate experiments larger surface cooling is simulated in the higher latitudes compared with the tropics which is consistent with the polar amplification simulated for an increase in $CO_2$ (Fig. S7a). Compared to other stratospheric sulfate experiments, the lower net negative radiative forcing in the Volc_100hPa case contributes to a global mean surface cooling of 2.18 K with respect to the 2XCO2 experiment, which attains only 70% of the 1XCO2 surface cooling. For the Volc_70hPa case, the surface cooling increases to 2.57 K relative to the 2XCO2 case reaching 82% of cooling due to halving of $CO_2$. Larger negative forcing in the Volc_37hPa case compared to other cases leads to more surface cooling in Volc_37hPa and thereby attaining ~93% of the cooling simulated in the 1XCO2 case.

The residual surface temperature patterns in the stratospheric aerosol simulations relative to 1XCO2 experiment (Fig. S7) shows a large warming at the higher latitudes. This is in agreement with several previous studies (Govindasamy et al., 2003; Kravitz et al., 2016; Nalam et al., 2017). The net surface cooling in the Volc_37hPa is less than the 1XCO2 experiment even-though the net negative radiative forcing surpasses the 1XCO2 radiative forcing (Fig. 1b and 3a). This is partly attributed to the lower efficacy of sulfate aerosol forcing (0.86-0.89; Sect. 3.2) and partly to the $CO_2$ physiological effect. For counteracting the global mean surface warming, the magnitude of sulfate forcing should be larger than $CO_2$ radiative forcing as its efficacy is less than one. The lower efficacy of sulfate aerosols is similar to the case of solar forcing which has an efficacy of about 0.8 (Schmidt et al., 2012; Modak et al., 2016; Duan et al., 2018). The $CO_2$ physiological forcing is caused by elevated $CO_2$ concentration, where the plant stoma opens less widely leading to less canopy transpiration and reduced evapotranspiration which leads to an increase in mean surface air temperature over land (Cao et al., 2010).

The surface cooling caused by a halving of $CO_2$ in 1XCO2 experiment is associated with a decrease in global mean precipitation by 5.8% relative to the 2XCO2 experiment (Fig. 3b and 5a). In the stratospheric aerosol experiments, even though the global mean surface cooling less than the 1XCO2 experiment, a larger reduction in global mean precipitation is simulated. Fast adjustments to $CO_2$ radiative forcing results in precipitation suppression (Bala et al., 2010; Ferraro et al., 2014), associated with an increase in stability in the lower troposphere (Bala et al., 2010; Cao et al., 2012). Thus, the fast response to a reduction





in $CO_2$ radiative forcing involves an increase in precipitation. In contrast, the fast response to the negative forcing from the introduction of a stratospheric aerosol layer does not involve an increase in global mean precipitation. For all types of forcing, a decrease in global mean temperatures is associated with decreases in global mean precipitation. For our volcanic aerosol simulations, both equilibrium global mean surface temperatures and equilibrium global mean precipitation decrease with

increasing altitude of the stratospheric aerosol layer (Fig. 3b). The spatial patterns (Fig. 5) show that the reduction in precipitation is larger over the tropics which is consistent with earlier studies (Govindasamy et al., 2003; Kravitz et al., 2013; Tilmes et al., 2013).

### 3.4 Stratospheric dynamics

An increase in atmospheric $CO_2$ concentration causes a warming of the surface and the troposphere but a cooling in the

stratosphere and mesosphere (Goessling and Bathiany, 2016 and references therein). Additionally, the local warming by aerosols can affect the dynamics of the stratosphere (Aquila et al., 2014; Niemeier and Schmidt, 2017).

We have performed a set of additional, 12-member ensemble simulations lasting 1 day to evaluate the effects of aerosols on the radiative heating. These 1-day runs provide an estimate of the instantaneous radiative effects of the prescribed aerosols. Monthly restarts from the 60th year of the prescribed SST control run are used to initialize these 1-day runs. Each

member of the ensemble starts from the first day of each calendar month (1st January, 1st February... etc.) and the simulation is performed with hourly outputs from the model. By averaging the 12 ensemble runs, the effects of seasonal cycle on the radiative forcing estimates are excluded.

The SW heating rate increases with the prescribed altitude, with a maximum warming of 0.24 K day$^{-1}$ for Volc_100hPa to 0.39 K day$^{-1}$ for Volc_70hPa, and 0.43 K day$^{-1}$ for Volc_37hPa (Fig. S8a). This is because the amount of solar

radiation decreases downward due to attenuation and hence more SW radiation is available at higher altitudes. For LW heating, a maximum of 0.32 K day$^{-1}$ is simulated for the Volc_70hPa case while the maximum is 0.13 K day$^{-1}$ for the Volc_100hPa case, and 0.16 K day$^{-1}$ for the Volc_37hPa (Fig. S8b). We are not aware of the reason for the maximum LW heating in the Volc_70hPa. Due to the differing SW and LW radiative heating rates in the three cases, the maximum heating rate and temperature change is simulated for the Volc_70hPa case (0.68 K day$^{-1}$), followed by Volc_37hPa (0.58 K day$^{-1}$), and

Volc_100hPa (0.34 K day$^{-1}$) (Fig. S8c).

To illustrate the effects of aerosol-induced warming on the dynamics of the stratosphere, changes in zonal mean temperature and wind for the stratospheric aerosol experiments with respect to the 2XCO2 simulation are analyzed (Fig. 6). Although the aerosols are prescribed uniformly around the globe, for the same altitude more warming is simulated in the tropics than in the poles as the mean incoming solar radiation is larger in the tropics. A maximum warming of approximately

6 K is simulated for the Volc_100hPa case relative to the 2XCO2 experiment. The maximum warming increases to almost 15 K for the Volc_70hPa case and it is approximately 10 K for the Volc_37hPa case. The uneven meridional radiative heating



(Fig. 6) alters the thermal wind balance in the stratosphere, and related changes in pressure gradients generate westerly wind anomalies (Ferraro et al., 2011; Aquila et al., 2014). Large wind anomalies are simulated for the Volc_70hPa and Volc_37hPa cases as the radiative heating was larger in these two cases (Fig. S8).

The radiative heating by sulfate aerosols, especially in the lower stratosphere leads to an increase in temperature of the tropical tropopause layer and an associated increase in water vapor transport from the troposphere to the stratosphere (Dessler et al., 2013). When the volcanic aerosols are prescribed at 100 hPa, the warming is near the tropical tropopause and it causes a significant increase in water vapor in the stratosphere (Fig. 7). An increase in specific humidity of 60% is simulated in the stratosphere for the Volc_100hPa case relative to the 2XCO2 experiment (Fig. 7). Though the radiative heating is largest for the Volc_70hPa experiment, the altitude of the layer is much above the tropopause and thus only a 25% increase in specific

humidity is simulated. No significant changes in specific humidity are simulated for Volc_37hPa case.

**3.5 Effects on Terrestrial vegetation productivity**

The vegetation primary productivity on land is proportional to the available photosynthetically active radiation at the surface (Pinker and Lazlo, 1992), which is the sum of direct and diffuse solar radiation (Alados and Alados-Arboledas, 1999). In our stratospheric aerosol experiments, a reduction in direct solar radiation reaching the surface is simulated due to the increased

SW scattering by the aerosols (Fig. S9, Table 2). However, the diffuse solar radiation reaching the surface increases (Fig. S9b; Kalidindi et al., 2015). This causes diffuse fertilization effect where the increase in diffuse radiation leads to increased productivity by increasing the light availability to a larger fraction of the canopy which otherwise remain shaded (Mercado et al., 2009; Kanniah et al., 2012). Thus, the diffuse radiation can cause an increase in productivity and can enhance the terrestrial carbon uptake (Alton et al., 2007; Mercado et al., 2009). Increased diffuse radiation availability and suppressed plant and soil

respiration due to cooling can enhance the terrestrial carbon sink in a sulfate geoengineering scenario (Xia et al., 2016).

The changes in the global mean values over land for the diffused and direct solar radiation components and corresponding changes in primary productivity for the experiments with respect to the 2XCO2 case are shown in Table 2. In the 2XCO2 experiment, there is an increase of 26.72 Gt-C yr$^{-1}$ (22.5%) in gross primary productivity (GPP) compared to the 1XCO2 experiment although the amount of radiation available for productivity is approximately the same in both cases. A

doubling of $CO_2$ concentration causes an increase in the plant productivity due to the $CO_2$ fertilization effect (Farquhar, 1997; Owensby et al., 1999). To exclude the $CO_2$ fertilization effect and assess the changes due to only the prescribed aerosols in the stratospheric aerosol experiments, changes in radiation and productivity are discussed relative to the 2XCO2 experiment below.

For the Volc_100hPa case, a decrease of 9.5 W m$^{-2}$ (-6.64%) in direct radiation relative to the 2XCO2 case is

simulated. The reduction in direct radiation increases with the altitude of aerosols to -11.4 W m$^{-2}$ (-8%) and -12.5 W m$^{-2}$ (-8.8%) for the Volc_70hPa, and Volc_37hPa cases, respectively. An increase in diffuse radiation of 8.1 W m$^{-2}$ is simulated for





the Volc_100hPa case, which is 18.7% larger than in the 2XCO2 case. As the height of volcanic aerosol increases, the increase in diffuse radiation at the surface becomes larger and the increase reaches 26% (11.2 W m$^{-2}$) of the 2XCO2 case for the Volc_37 hPa case. The changes simulated in the diffuse radiation and direct radiation are of similar magnitude in all sulfate aerosol experiments. Thus, the decrease in the direct radiation is partially offset by the increase in availability of diffuse radiation at

the surface. From Table 2, it can be inferred that the net reduction in solar radiation at the surface is about 1.3 W m$^{-2}$ (~0.7%).

Spatial patterns of direct and diffuse radiation change relative to 2XCO2 experiment for the Volc_100hPa case over the land shows that overall there is a decrease in direct radiation and an increase in diffuse radiation all over the globe (Fig. S9a, b). We found that these patterns are similar for the other two stratospheric sulfate experiments. Large changes in direct and diffuse radiation are simulated in the dry regions and desserts. Spatial pattern of vertically integrated cloud cover in the

2XCO2 case (Fig. S9c) show that these large changes in direct and diffuse solar radiation occur in areas where the total cloudiness is small.

The total GPP is the sum of sunlit GPP (which depends on direct solar radiation) and shaded GPP (which depends on diffuse solar radiation). The changes in total GPP in all cases are dominated by the change in sunlit GPP (Table 2). Although the changes in direct and diffuse radiations are comparatively of similar magnitudes, the decrease in sunlit GPP is significantly

more (by an order of magnitude) than the increase in shaded GPP (Table 2). Thus, the additional productivity due to the increased diffused radiation availability is overwhelmed by the reduction in sunlit GPP. Other studies have also found that the effect from reduced direct radiation dominate the effect of increased diffuse radiation, and thus the net effect of sulfate geoengineering is to reduce plant productivity (e.g. Kalidindi et al., 2015).

The decrease in sunlit GPP is less when volcanic aerosols are prescribed at the lower levels of the stratosphere as the

reduction in direct sunlight at the surface is less. The changes in GPP can also be modulated by the availability of nitrogen as simulated by the CN (carbon and nitrogen) module in CLM4. When aerosols are prescribed at lower levels, there is less cooling which causes relatively more mineralization reducing the nitrogen limitation effect (Rustard et al., 2001). The net primary productivity (NPP) shows similar changes (Table 2): the minimum decrease in NPP is simulated in the Volc_100hPa case (-0.96 Gt-C yr$^{-1}$ relative to the 2XCO2 case). The percentage decrease in NPP (2.0 to 3.5%) is smaller than GPP (5.0 to 7.6%)

because of a decrease in autotropic respiration in the stratospheric aerosol experiments (NPP equals GPP minus autotropic respiration; Table 2). The decrease in autotropic and heterotrophic respiration is related to a relatively cooler climate in the stratospheric aerosol experiments compared to the 2XCO2 case.

## 4 Discussion and conclusion

Sensitivity of radiative forcing and surface temperature to the altitude of volcanic size sulfate aerosols in the stratosphere is

analyzed in this study using a climate model with prescribed aerosol distributions. The model used is less comprehensive than models which simulate the aerosol microphysics, transport and removal processes. By excluding these processes, we isolate



the dependence of radiative forcing on the height of the aerosol layer. The sensitivity experiments are performed by prescribing aerosols of a size characteristic of volcanoes at three different altitudes in the stratosphere (100 hPa, 70 hPa, and 37 hPa) but keeping the total mass of the volcanic aerosols constant at 20 Mt (15 Mt of $H_2SO_4$).

We show that for the same additional aerosol mass, volcanic aerosols produce more negative radiative forcing when they are prescribed at higher altitudes in the stratosphere (Fig. 1b). Since the microphysical or transport processes is not included in this study, the global mean surface temperature change is solely dependent on radiative forcing which is sensitive to the prescribed altitude of aerosols. The radiative heating by volcanic aerosol in the lower stratosphere leads to increased stability of the troposphere and a reduction in the high cloud cover by the "cloud-burn-off" effect and increased water vapor transport to the stratosphere (Fig. S2; Visioni et al., 2018). However, the resulting negative LW forcing from cloud cover
change is overwhelmed by the large positive LW forcing due to the absorption of radiation by prescribed aerosols (Figs. 2d, 2f). The high cloud changes are sensitive to the proximity of the heated layer to tropopause and is thus sensitive to the altitude of the aerosols. The changes in tropospheric stability also contributes to changes in low cloud cover (Fig. S3). Further, the changes in tropopause cold point temperature due to the radiative warming of the lower stratosphere and increased stratospheric humidity affects the clear-sky radiative forcing. The positive LW forcing offsets a part of the negative SW forcing in the
stratospheric aerosol experiments. Thus, our study also highlights the importance of LW forcing in the efficiency of the stratospheric aerosol experiments and the need for carefully accounting LW forcing along with the SW forcing (Kleinschmitt et al., 2018).

    The differences simulated in radiative forcing are reflected in the surface temperature response and we find that volcanic aerosols cause more surface cooling when they are prescribed at higher levels of the stratosphere. Assuming a lifetime
of 1 year of the aerosols in the stratosphere, the 20 Mt of aerosol used in this study is equivalent to 9.79 Tg yr$^{-1}$ $SO_2$ injection (or 4.9 Tg-S yr$^{-1}$). As this amount at 37hPa almost completely attains the halving of $CO_2$ induced cooling, the efficiency in cooling the surface is estimated as 0.59 K Tg-S$^{-1}$. The corresponding efficiencies for the Volc_70hPa and Volc_100hPa simulations are 0.52 K Tg-S$^{-1}$ and 0.44 K Tg-S$^{-1}$, respectively. The surface temperature difference between our stratospheric aerosol experiments shows that even when the processes such as aerosol microphysics, transport and sedimentation are
excluded, the differences in effective radiative forcing between the stratospheric aerosol simulations experiments is substantial.

    For 6 Tg-$SO_2$ yr$^{-1}$ injections, Tilmes et al. (2017) estimated a cooling of ~0.22 K Tg-S$^{-1}$ for equatorial high-altitude injections (30 hPa) and ~0.18 K Tg-S$^{-1}$ for equatorial low-altitude injections (60 hPa) when aerosols concentrations in the stratosphere had reached a steady state. While our results agree in sign, it should be noted that while Tilmes et al. (2017) estimated efficiency from experiments with sulfur emissions we have made estimates using prescribed aerosol burden. Further,
the surface cooling discussed in Tilmes et al. (2017) are from 10-year coupled simulations where the climate system has not reached a steady state, while our results are from equilibrium simulations. Further differences can be attributed to differing model configurations (slab ocean versus fully coupled) and different versions of the model used in the two studies.



With the surface cooling in stratosphere aerosol experiments, a reduction in global mean precipitation is simulated in the stratospheric aerosol experiments as shown in several previous studies (Bala et al., 2010; Modak and Bala, 2014; Nalam et al., 2017). The reduction in global annual mean precipitation increases as the height of the aerosol layer increases. Because of the absorption of radiation by volcanic aerosols, a significant warming in the stratosphere is simulated as reported in many

previous studies (Ferrarro et al., 2011; Niemier and Schmidt, 2017; Richter et al., 2017). The magnitude of radiative warming is also sensitive to the altitude of the aerosols and a maximum warming of 15 K is simulated relative to the 2XCO2 experiment for the case where aerosols are prescribed at 70hPa. The maximum warming simulated here is comparable to the maximum warming of 10 to 15 K simulated in other studies such as Richter et al. (2017) and Tilmes et al. (2018b). The aerosol induced stratospheric warming and the resulting strong stratospheric westerly wind anomalies are sensitive to the altitude of the

aerosols. Further, the radiative heating in the lower stratosphere causes the tropical upper tropopause layer to warm which leads to increased water vapor transport into the stratosphere. In the stratospheric aerosol experiments, due to shortwave scattering by aerosols, there is an increase in diffused solar radiation and a decrease in direct solar radiation reaching the surface. Correspondingly, an increase in shaded GPP and a decrease in sunlit GPP are simulated. The net result is a decrease in GPP in all cases as the decrease in sunlit GPP is significantly larger compared to the increase in shaded GPP.

There are several limitations to this study. First, the aerosols are distributed uniformly at specific heights with fixed particle size distributions in our simulations, which is likely not achievable in an actual stratospheric sulfate deployment scenario. Our experiments do not include the effects of particle growth, aerosol chemistry, transport as well as its removal processes. The volcanic aerosol geometric mean radius used here (0.423 µm) is very close to the size where significant sedimentation can occur (Tilmes and Mills, 2014). A lack of ozone chemistry in the model and the absence of events such as

internally generated quasi-biennial oscillation (QBO) limits detailed analysis on stratospheric responses to the radiative warming by aerosols (Aquila et al., 2014; Kleinschmitt et al., 2018). For computational efficiency, we have used slab ocean version of the coupled model instead of fully dynamic ocean component and hence the transient effects and deep ocean feedbacks are missing in our study. Despite these limitations, we believe that our conclusions on the dependence of the radiative forcing and hence the surface climate on the altitude of aerosol layer are more fundamental and the robustness of our results

should be assessed using multiple models in a future study.

To summarize, for the same mass, the efficiency (defined as changes in surface temperature per Tg-S) of volcanic aerosol is less when they are prescribed at the lower altitudes in the stratosphere (Fig. 8). For example, in our simulations, there is a surface cooling of 0.44 K for each Tg-S placed in the stratosphere at about 16 km altitude (100 hPa). There is an additional surface cooling of 0.15 K per Tg-S when the prescribed altitude is increased from about 16 km to about 22 km (37

hPa).



*Code and data availability*. Model outputs and analysis scripts are available on request from the corresponding author.

*Author contributions*. KKS and GB designed the study, analyzed the data and wrote the manuscript. LC, LD, and KC contributed to the study design and writing of the manuscript.

*Competing interests*. Author GB is a member of the editorial board of the journal. Other authors declare that they have no
conflict of interest.

*Acknowledgements*. This work was funded by the Department of Science and Technology (DST), India grant number DST/CCP/MRDP/96/2017(G). Numerical simulations were performed on the supercomputer Sahasrat at the Supercomputer Education and Research Centre (SERC), Indian Institute of Science, Bangalore. The authors thank Angshuman Modak (Indian Institute of Science) and Adithya Nalam (Institute for Advanced Sustainability Studies Potsdam) for their technical help.

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



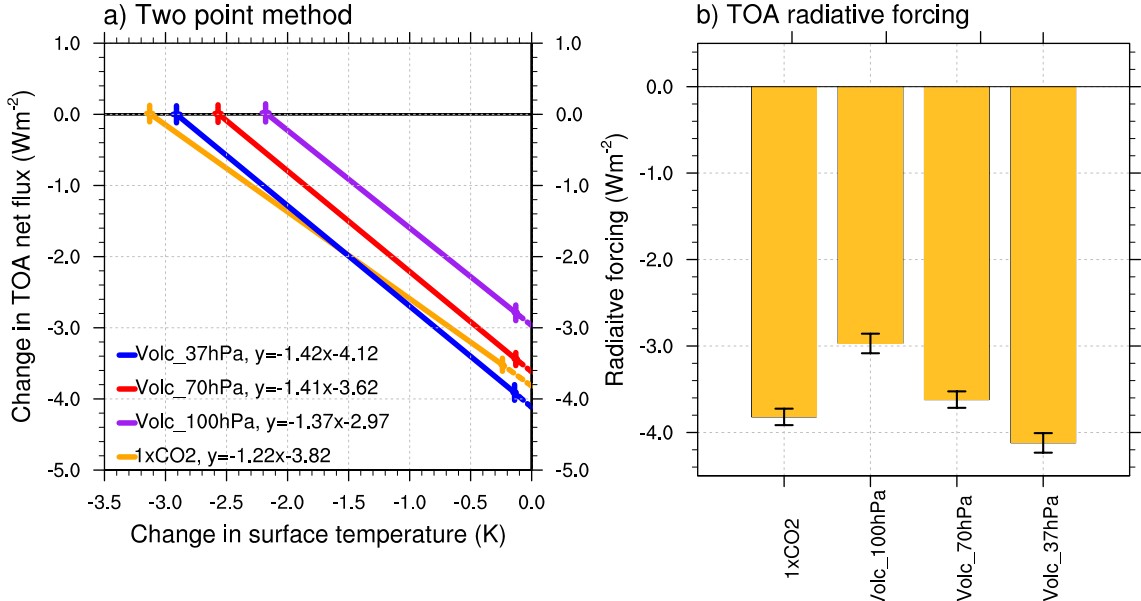

**Figure 1:** (a) The two-point method for estimating radiative forcing and feedback parameter (Supplementary Sect. S1). Change in global and annual mean surface temperature and TOA radiative imbalance from the slab ocean (points on the left) and prescribed SST (points on the right) simulations relative to the 2XCO2 simulation. The climate feedback parameter (slope of the lines) and the effective radiative forcing (intercept on the y-axis on the right) for $CO_2$ change (1XCO2-2XCO2) and all stratospheric sulfate experiments can be inferred from the linear regression relationships shown in the figure legends. Horizontal and vertical bars show 2 standard errors of the annual mean differences in surface temperature and radiative forcing relative to 2XCO2 experiment, respectively. The standard errors are estimated using 30 annual means for prescribed-SST simulations and 60 annual means for slab ocean simulations. (b) The global annual mean TOA radiative forcing at top of the atmosphere relative to the 2XCO2 experiment, estimated using the two-point method as illustrated in panel (a). The error bars represent 2 standard errors calculated from 30 annual means of the difference from the 2XCO2 experiment.





**Figure 2:** TOA SW and LW radiative forcing for all-sky (top panels) clear-sky (middle panels) and cloudy-sky (bottom panels) conditions relative to the 2XCO2 experiment, estimated using the two-point method (Supplementary Sect. S1). The error bars represent 2 standard errors calculated from 30 annual means of the difference from the 2XCO2 experiment.





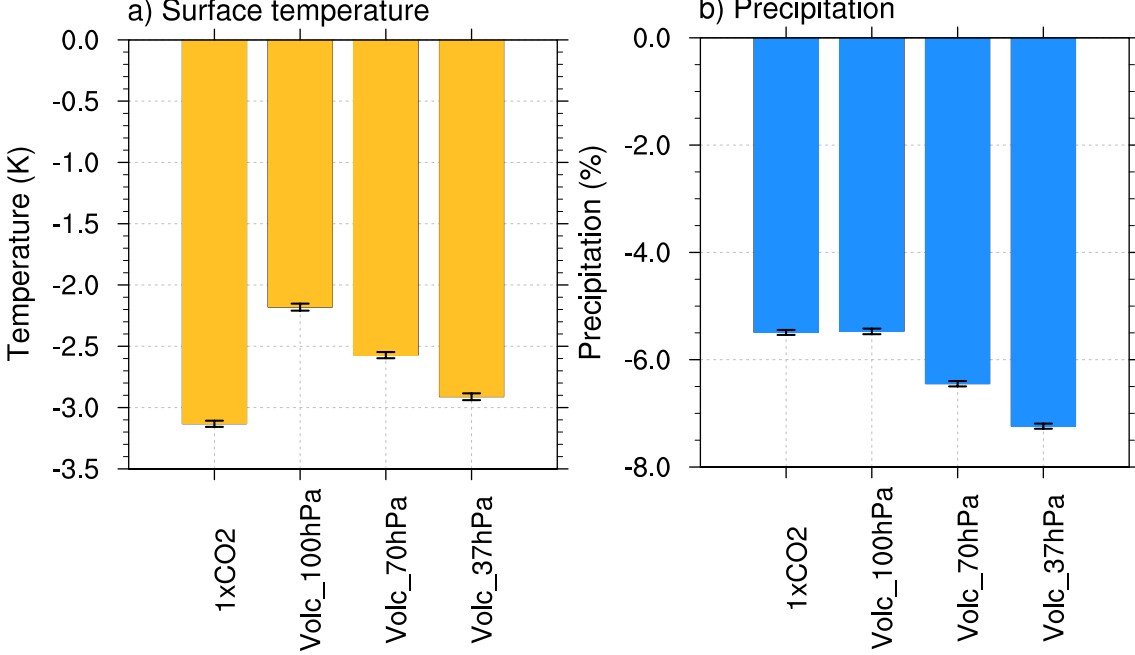

**Figure 3:** Changes in global and annual mean (a) surface temperature and (b) precipitation relative to the 2XCO2 experiment (slab ocean simulations). The error bars represent 2 standard errors calculated from 60 annual means of the difference from the 2XCO2 experiment.

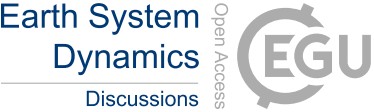



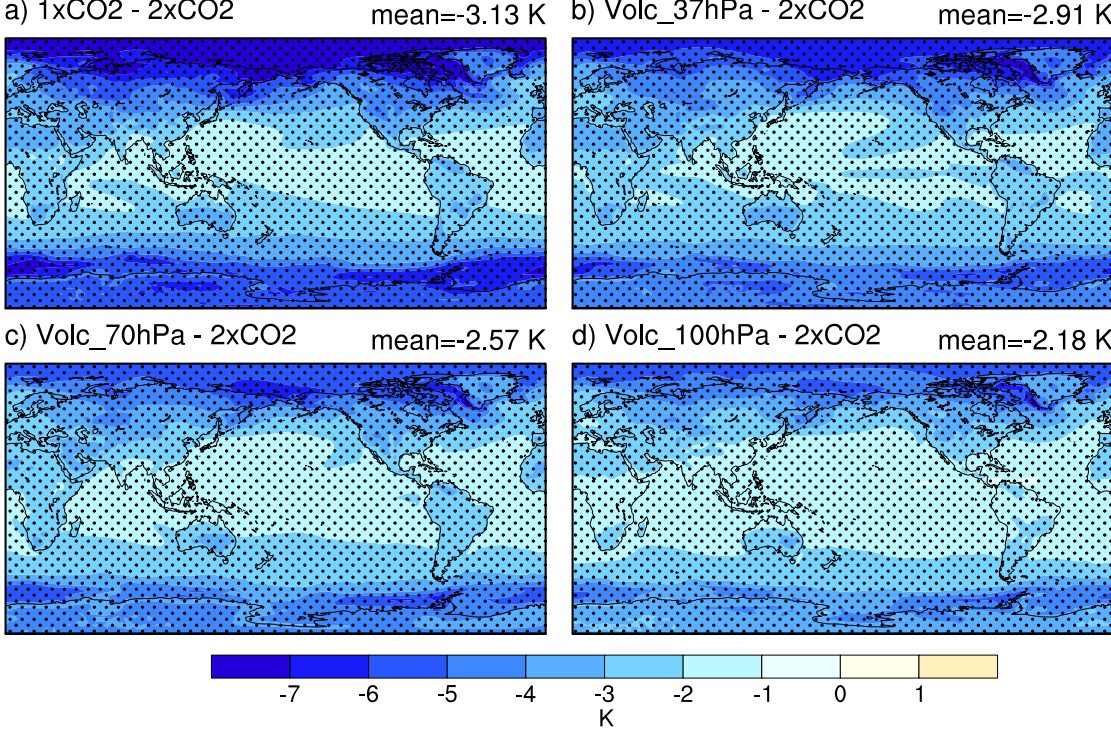

**Figure 4:** The spatial pattern of changes in surface temperature relative to the 2XCO2 experiment (slab ocean simulations). The hatched areas show the regions where the changes are significant at the 5% significance level. Significance level is estimated using Students t-test from 60 annual means of the experiments. Global mean value of the changes in each experiment is shown at the top right of each panel.

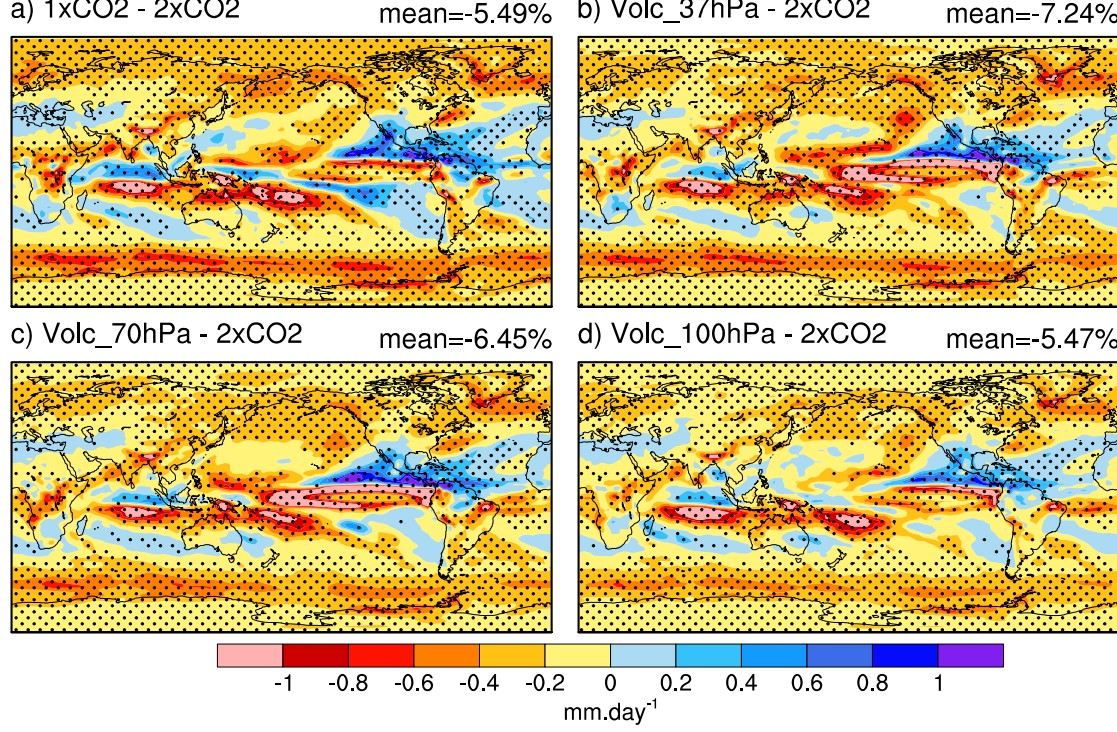

**Figure 5:** The spatial pattern of changes in precipitation relative to the 2XCO2 experiment (slab ocean simulations). The hatched areas show the regions where the changes are significant at the 5% significance level. Significance level is estimated using Students t-test from 60 annual means of the experiments. Global mean value of the changes in each experiment is shown at the top right of each panel.

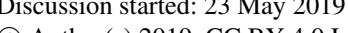



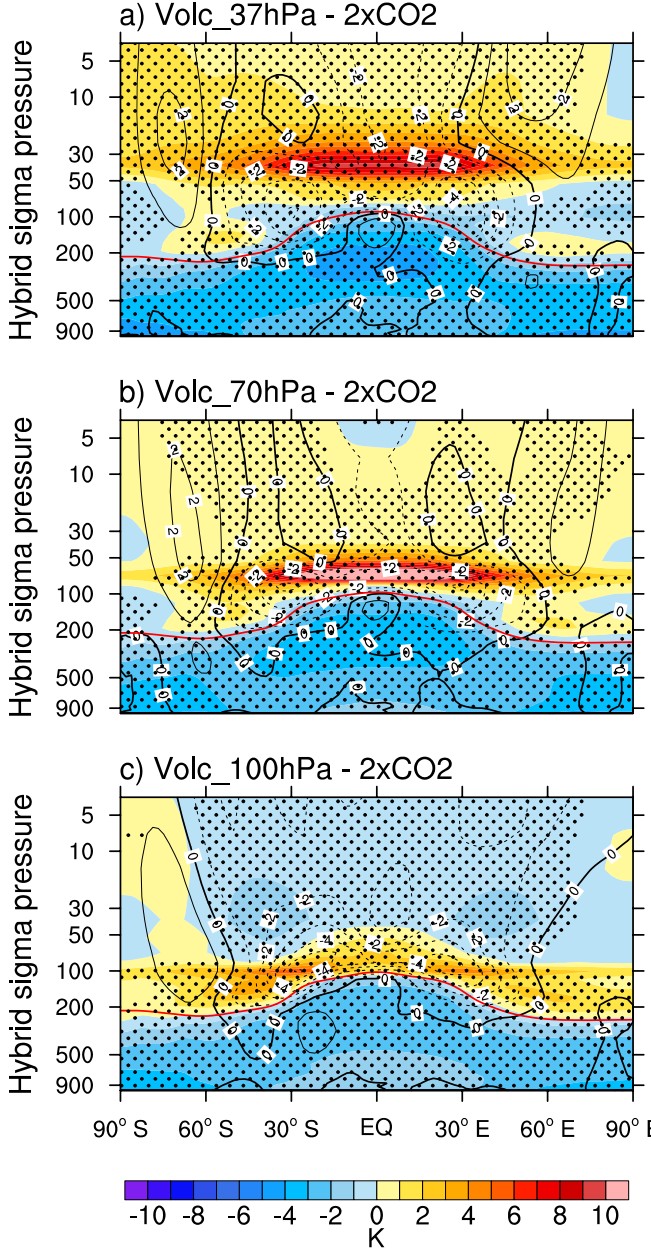

**Figure 6:** The changes in zonal average temperature (shaded) and winds (contours) in the three stratospheric sulfate simulations relative to the 2XCO2 simulation (slab ocean simulations). Position of the tropopause in each case is marked as a red line. The hatched areas in the plot show the regions where the changes are significant at the 5% significance level. Significance level is estimated using Students t-test from 60 annual means of the experiments.





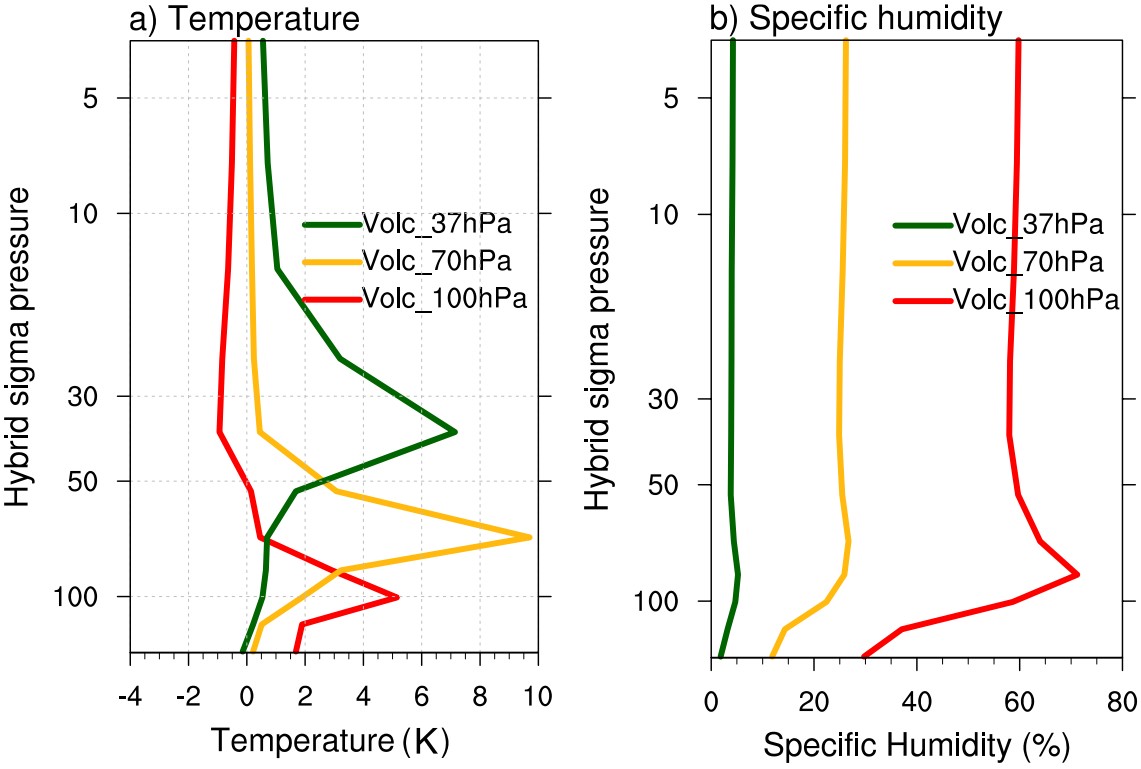

**Figure 7:** Vertical profiles of changes in global and annual mean of stratospheric (a) temperature and (b) specific humidity in percentage for the stratospheric sulfate simulations relative to the 2XCO2 experiment (slab ocean simulations). Lines are linear interpolations between layer midpoint values.




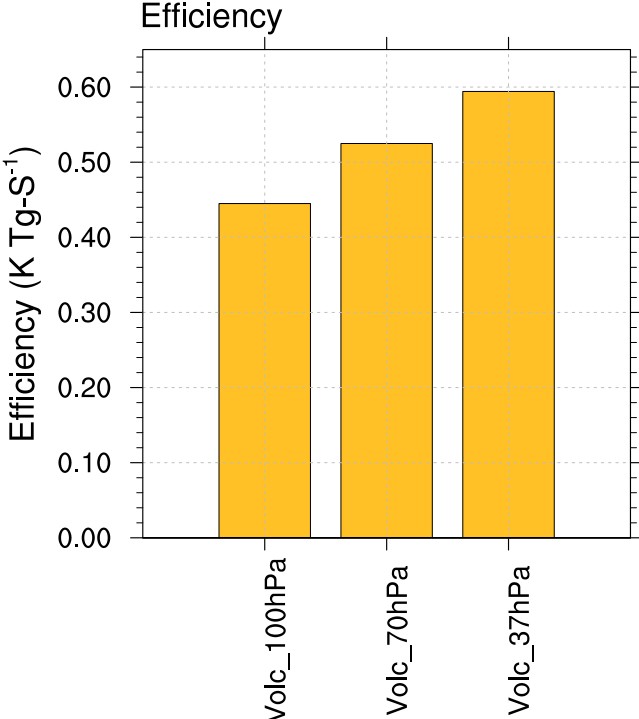

**Figure 8:** Change in global mean surface temperature per Tg-S in the stratosphere (efficiency) in the volcanic aerosol simulations.



**Table 1.** Radiative forcing estimates and global annual mean temperature changes relative to the 2XCO2 case, and climate sensitivity and efficacy. Uncertainties for changes are estimated as the 2 standard errors. Uncertainties are calculated from 60 annual mean differences for slab ocean and 30 annual means for prescribed SST experiments. The two-point method of estimating the radiative forcing, climate sensitivity and efficacy are discussed in Supplementary Sect. S1.

| | 1XCO2 | Volc_100hPa | Volc_70hPa | Volc_37hPa |
|---|---|---|---|---|
| Radiative forcing (Prescribed-SST method; W m$^{-2}$) | -3.52±0.09 | -2.79±0.11 | -3.44±0.09 | -3.91±0.11 |
| Global mean temperature change (K) | -3.13±0.03 | -2.18±0.03 | -2.57±0.03 | -2.91±0.03 |
| Radiative forcing (two-point method; W m$^{-2}$) | -3.82±0.09 | -2.97±0.11 | -3.62±0.26 | -4.2±0.11 |
| Climate feedback parameter (two-point method; W m$^{-2}$ K$^{-1}$) | 1.22±0.05 | 1.37±0.09 | 1.41±0.07 | 1.42±0.06 |
| Efficacy relative to CO$_2$ forcing (two-point method) | One | 0.89±0.07 | 0.87±0.05 | 0.86±0.05 |





**Table 2.** Global and annual mean values of key land model variables from the 1XCO2 and 2XCO2 simulations and the change in these variables in the stratospheric sulfate experiments relative to the 2XCO2 experiment. Uncertainties for changes are estimated as 2 standard errors calculated from 60 annual mean differences. Uncertainties for 1XCO2 and 2XCO2 cases are estimated as the standard deviation from the 60 annual means. Percentage changes from the 2XCO2 simulation is given in parenthesis.

|  | 1XCO2 | 2XCO2 | Volc_100hPa minus 2XCO2 | Volc_70hPa minus 2XCO2 | Volc_37hPa minus 2XCO2 |
|---|---|---|---|---|---|
| Diffuse radiation (W m$^{-2}$) | 44.63±0.13 | 43.23±0.17 | 8.09±0.09 (18.7%) | 9.91±0.06 (22.9%) | 11.22±0.05 (26%) |
| Direct radiation (W m$^{-2}$) | 142.73±0.51 | 142.23±0.18 | -9.45±0.22 (-6.6%) | -11.37±0.21 (-8%) | -12.50±0.23 (-8.8%) |
| Shaded GPP (Gt-C yr$^{-1}$) | 56.66±0.58 | 63.62±0.53 | 0.28±0.20 (0.4%) | 0.61±0.22 (1%) | 0.50±0.28 (0.8%) |
| Sunlit GPP (Gt-C yr$^{-1}$) | 62.17±0.48 | 81.93±0.67 | -7.52±0.29 (-9.2%) | -10.09±0.25 (-12.3%) | -11.63±0.28 (-14.2%) |
| GPP (Gt-C yr$^{-1}$) | 118.83±0.96 | 145.55±1.15 | -7.23±0.45 (-5%) | -9.48±0.45 (-6.5%) | -11.13±0.53 (-7.6%) |
| NPP (Gt-C yr$^{-1}$) | 41.36±0.44 | 47.90±0.55 | -0.96±0.19 (-2%) | -1.33±0.19 (-2.8%) | -1.69±0.22 (-3.5%) |
| Autotrophic Resp. (Gt-C yr$^{-1}$) | 77.47±0.74 | 97.65±0.92 | -6.27±0.35 (-6.43%) | -8.16±0.33 (-8.36%) | -9.44±0.41 (-9.67%) |
| Heterotrophic Resp. (Gt-C yr$^{-1}$) | 39.01±0.25 | 44.54±0.38 | -1.14±0.13 (-2.56%) | -1.48±0.12 (-3.31%) | -1.76±0.14 (-3.96%) |
| Vegetation carbon (Gt-C yr$^{-1}$) | 596.64±2.87 | 706.88±7.05 | -3.54±1.40 (-0.50%) | -5.57±1.47 (-0.79%) | -7.68±1.34 (-1.09%) |
| Soll carbon (Gt-C) | 471.81±0.19 | 470.91±1.34 | 12.65±0.65 (2.69%) | 15.01±0.79 (3.19%) | 16.44±0.85 (3.49%) |
| Total ecosystem carbon (Gt-C) | 1068.45±2.93 | 1177.79±8.31 | 9.11±2.02 (0.77%) | 9.43±2.25 (0.80%) | 8.76±2.12 (0.74%) |