# Peer review of "Climate System Response to Stratospheric Sulfate Aerosols: Sensitivity to Altitude of Aerosol Layer"

_Earth System Dynamics, 2019_

## Referee Comment (RC1) · Anonymous Referee #1 · 1 Jun 2019

I reviewed an earlier version of this manuscript that was submitted to another journal. I think this version is far improved over the previous one. Many of the results are perhaps not all that surprising, especially given that other studies (which the authors cite) have looked at the different effects of injection altitude. However, I have not necessarily seen all of these results in one place, which makes this paper interesting. The addition of Section 3.5 is very interesting, and I learned quite a bit. I am recommending just a few minor revisions.

General comments:

As the authors say, they don't include any dynamics or transport. However, radiative

forcing and climate response to stratospheric aerosols definitely depend on dynamics. I would appreciate the authors adding some description as to how this compromise might have affected their results.

Specific comments:

In the last paragraph on Page 5 (going into Page 6), some context for these results is needed. Do these numbers make sense, and why? (I think they make sense, but I'd like for you to say so.)

Page 8, line 13: Can you phrase this in a different way? 1xCO2 is your baseline, so it doesn't cause any cooling.

Figure 4: I'm not sure hatching is necessary. All of the regions are statistically significant, so just say that.

---

## Referee Comment (RC2) · Anonymous Referee #2 · 13 Jun 2019

The authors simulate the solar radiation method (SRM) of stratospheric sulfate aerosols by prescribing a uniform layer of sulfate aerosol concentration. Assuming different altitudes of this layer, they determine the impact on different variables like radiative forcing, surface temperature, and humidity and temperature in the stratosphere.

The paper is very well written and includes an impressive literature review. I recommend publication after the authors addressed the following comments.

**General comments**

My main concern is related to the prescribed aerosol layer. Prescribing aerosols in a climate model is a widely used technique and can be used to be computationally more

efficient. However, the prescribed layer should be comparable to reality and preferable been calculated previously with a aerosol microphysical model. This study simplified the method by prescribing the aerosol concentration meridionally uniformly at a certain height (km) above sea level. This distribution is quite unrealistic. However, this may not have extremely strong impact on the main conclusion of the paper, the dependency of radiative forcing on altitude.

Stronger impact has the fact that the prescribed profiles do not include the changes caused by sedimentation. Sedimentation causes a vertical spread of the aerosols, the stronger the higher the injection altitude above the tropopause (e.g. (Tilmes et al. (2017)). This difference to a more realistic profile is not mentioned in the paper. As a consequence, heating of the aerosols will not occur in a small layer as shown in Figure 6a and 7. The heated layer extends to the tropopause. This has consequences for clouds and humidity transport. This aspect is not at all discussed in the paper.

I agree with the authors that a simplification is useful and can help to gain new insight. The dependency of the radiative forcing on the altitude of the SRM agent has been shown for three different SRM techniques by Niemeier et al (2013). Sulfur injection studies have shown that the resulting radiative forcing depends on the altitude of the injection. This has mostly been related to the vertical extension of the aerosol layer. This paper could give new insights by adding a simulation with more realistic sulfate profile. This would show whether the altitude of the injection or the vertical extension of the aerosol layer is the cause of an increased radiative forcing.

The limitation of the model, no aerosol microphysics, are well described. The more important limitation of the prescribed sulfate concentration are not prescribed. Please, add a figure of the prescribed sulfate layers in the main article. For a reader who is not familiar with stratospheric sulfate distribution it would be very helpful.

**Specific comments**

Page 3 Line 27: I don't understand this sentence.
Section 2.2:

How is the sulfate layer created? The global distribution in Nalam et al (2017) is unrealistic. Aerosols follow tropopause in a meridional cross section (e.g. Visioni et al. (2017) or Dhomse et al. (2014)).

Add figures of sulfate distribution.

It remains unclear how the sulfate is distributed vertically. I assume you only change the height of the prescribed layer.

Page 4 Line 22: 20 Mt SO4? Change unit to Tg (SI unit)

Page 6 Line16: Thus, CO2 reduction would be most effective. This would be a good statement in the conclusion.

Page 7 Line 16 - 26: Results of Kuebbeler et al (2012) should be taken into account.

Page 8 Line 30: See also Liepert and Prevedi (2009) and Kravitz et al (2013).

Page 8 Line 27pp: This is a very short summary of this topic.

Page 9 Stratospheric dynamics: The vertical resolution of you model is very limited. Good results of stratospheric dynamics need a higher resolution, even in a model that is still not capable to generate a QBO. You should mention the QBO in the first paragraph and not only cite the two papers.

Page 9 Line 27: You show significance for the temperature. How about the zonal wind? Your differences are very small. Please add a figure with significance of the zonal wind.

Page 9 Line 29: Higher temperature in the tropics because of more solar radiation. The aerosols cannot absorb as much radiation during polar winter then during summer.

Page 10 Line 3: Significant?

Page 10 Line 4 to 10: Your vertical profile of aerosols is unrealistic. As a consequence the vertical profile of temperature anomaly and humidity anomaly are unrealistic. You
have to say this and the consequences for your results.

Page 12 Line 5 to 6: I don't understand this sentence? What would be the relation between microphysics and global mean surface temperature?

Page 12 Line 15-16: This is not a new result and can be found in many previous studies. I don't think there is a tendency to include radiation to aerosol coupling for SW only in the models, at least not in general.

Page 12 Line 25 Many studies show that higher altitude of injection causes stronger forcing. Most studies relate this to the thicker aerosol layer. Niemeier et al. (2013) have show that for different SRM methods the forcing depends on the altitude of the forcing agent. Thus, your result is not new. It remains open, whether the top level of the aerosols or the vertical extension of the sulfate layer is more important. Your study would gain a lot if you could answer this by adding a simulation with a more realistic vertical profile.

Page 13 Line 15 to 25: As stated above, the main limitation is the profile itself. The profile changes when injecting at higher altitude and the particle sediment. This has to be reflected in your profile and would change e.g. vertical humidity transport.

**References**

Dhomse, S. S., Emmerson, K. M., Mann, G. W., Bellouin, N., Carslaw, K. S., Chipperfield, M. P., Hommel, R., Abraham, N. L., Telford, P., Braesicke, P., Dalvi, M., Johnson, C. E., O'Connor, F., Morgenstern, O., Pyle, J. A., Deshler, T., Zawodny, J. M., and Thomason, L. W.: Aerosol microphysics simulations of the Mt. Pinatubo eruption with the UM-UKCA composition-climate model, Atmospheric Chemistry and Physics, 14, 11221–11246, doi:10.5194/acp-14-11221-2014, 2014

Kravitz, B., Rasch, P. J., Forster, P. M., Andrews, T., Cole, J. N. S., Irvine, P. J., Ji, D., Kristjánsson, J. E., Moore, J. C., Muri, H., Niemeier, U., Robock, A., Singh, B., Tilmes, S., Watanabe, S., and Yoon, J.-H.: An energetic perspective on hydrological cycle
changes in the Geoengineering Model Intercomparison Project (GeoMIP), J. Geophys. Res., 118, 13087–13102, doi: 10.1002/2013JD020502, 2013.

Kuebbeler, M., Lohmann, U., and Feichter, J.: Effects of stratopheric sulfate aerosol geo-engineering on cirrus clouds, Geophys. Res. Lett., 39, L23803, doi:10.1029/2012GL053797, 2012.

Liepert, B. G. and Previdi, M.: Do Models and Observations Disagree on the Rainfall Response to Global Warming?, J. Climate, 22, 3156–3166, doi: http://dx.doi.org/10.1175/2008JCLI2472.1, 2009.

Niemeier, U., Schmidt, H., Alterskjær, K., and Kristjánsson, J. E.: Solar irradiance reduction via climate engineering – Impact of different techniques on the energy balance and the hydrological cycle, JGR, 118, 11905–11917, doi:10.1002/2013JD020445, 2013.

Tilmes, S., Richter, J. H., Mills, M. J., Kravitz, B., MacMartin, D. G., Vitt, F., Tribbia, J. J., and Lamarque, J.-F.: Sensitivity of Aerosol Distribution and Climate Response to Stratospheric SO2 Injection Locations, Journal of Geophysical Research: Atmospheres, 122, 12,591–12,615, doi:10.1002/2017JD026888, 2017.

Visioni, D., Pitari, G., and Aquila, V.: Sulfate geoengineering: a review of the factors controlling the needed injection of sulfur dioxide, Atmospheric Chemistry and Physics, 17, 3879–3889, doi: 10.5194/acp-17-3879-2017, 2017

**ESDD**

---

## Referee Comment (RC3) · Anonymous Referee #3 · 14 Jun 2019

Review of

"Climate system response to stratospheric sulfate aerosols: sensitivity to altitude of aerosol layer" by Krishnamohan Krishna-Pillai Sukumara-Pillai, Govindasamy Bala, Long Cao, Lei Duan and Ken Caldeira.

General Comments

This is a well-structured paper which presents its results clearly, is well written with clear figures. The dependence of the amount of surface cooling on the altitude of the aerosol layer has been shown before (e.g. the work of Tilmes et al. [2017] referred to by the authors) so this work falls into the category of "confirmatory" rather than

"groundbreaking" work. My main concern relates to the ability of their model to simulate stratospheric dynamics well enough to have confidence in their results - see Specific Comment 1.

Specific Comments

1. Page 4, Section 2.1, with implications throughout. With a top at 3 hPa (c. 40 km) and 26 layers in the vertical the model is both "low top" and "low vertical resolution". This leads to concerns about how well the model represents stratospheric dynamics and therefore how much confidence can be had in any results based on such dynamics, such as the amount of water vapor entering the stratosphere (page 7, lines 5-11; page 10, lines 5-10; page 12, lines 12-14) and changes to stratospheric circulation (the whole of Section 3.4). It is not surprising that, as the authors admit, their model does not produce an internally-generated QBO, but one is left wondering how well the model simulates the Brewer-Dobson circulation. Some validation of the model's Brewer-Dobson circulation against observations is required in order to justify confidence in the results.

2. Page 4, lines 26-27. The manuscript at present simply states "The zonal variations as well as interannual variations (for this study) in mixing ratio of the volcanic aerosols are ommitted". Although they do make this clearer later in the Discussion/Conclusion, it needs to be made much clearer here that this means that their model includes no aerosol transport, deposition, microphysics or chemistry - that the aerosol layers are simply represented by fixed, globally-uniform values.

3. Page 7, lines 18-19. What the authors call the "burn-off effect" with reference to Ackerman et al. (2000) is completely irrelevant as an explanation here. Ackerman et al. examined the impact on boundary-layer trade cumuli of low-level soot. This has no bearing on the reduction of upper-tropospheric cirrus cloud being discussed at this point.

4. Page 12, lines 8-9. The authors again use the term "burn-off effect" but this time with

reference to Visioni et al. (2018). The term again seems inappropriate as Visioni et al. explain the thinning of high-altitude cirrus clouds in terms of an increase in atmospheric stability and thus a decrease in turbulence and updraft velocities - nothing about "burn-off".

Technical Corrections/Comments

1. Page 4, line 13: the number of model layers in the stratosphere should be given.

2. Page 7, line 22-23: the text currently reads "...leads to an increase in low cloud for the Volc_100hPa case relative to the Vol_70hPa and Volc_35hPa cases..." This is not incorrect, but I think it would be clearer to say "...leads to less of a decrease in low cloud for the Volc_100hPa case compared with the Volc_70hPa and Volc_35hPa cases..."

3. Page 11, lines 25-26: "autotrophic" is misspelled as "autotropic" three times.

4. Supplementary material, page 6: the caption to Figure S1 should explain what is shown in each of the panels (a) to (f).

5. Supplementary material, page 12: the term "1XCO2" is used in the caption to Figure S7 and has been used throughout the paper, but "CTL" is used in the titles of the individual panels; consistency would avoid any confusion.

6. Supplementary material, page 13: the values plotted in Figure S8 are presumably global-means?

---

## Author Comment (AC1) · 2 Aug 2019

Please find attached our reply to the comments of Reviewer-1

Please also note the supplement to this comment:
https://www.earth-syst-dynam-discuss.net/esd-2019-21/esd-2019-21-AC1-supplement.pdf

---

## Author Comment (AC3) · 2 Aug 2019

**Reply to the comments from Anonymous Reviewer-3**

Review of "Climate system response to stratospheric sulfate aerosols: sensitivity to altitude of aerosol layer" by Krishnamohan Krishna-Pillai Sukumara-Pillai, Govindasamy Bala, Long Cao, Lei Duan and Ken Caldeira.

**General Comments**

This is a well-structured paper which presents its results clearly, is well written with clear figures. The dependence of the amount of surface cooling on the altitude of the aerosol layer has been shown before (e.g. the work of Tilmes et al. [2017] referred to by the authors) so this work falls into the category of "confirmatory" rather than "groundbreaking" work. My main concern relates to the ability of their model to simulate stratospheric dynamics well enough to have confidence in their results - see Specific Comment 1.

We thank the reviewer for the constructive comments which helped us to further improve the manuscript.

**Specific Comments**

1. Page 4, Section 2.1, with implications throughout. With a top at 3 hPa (c. 40 km) and 26 layers in the vertical the model is both "low top" and "low vertical resolution". This leads to concerns about how well the model represents stratospheric dynamics and therefore how much confidence can be had in any results based on such dynamics, such as the amount of water vapor entering the stratosphere (page 7, lines 5-11; page 10, lines 5-10; page 12, lines 12-14) and changes to stratospheric circulation (the whole of Section 3.4). It is not surprising that, as the authors admit, their model does not produce an internally-generated QBO, but one is left wondering how well the model simulates the Brewer-Dobson circulation. Some validation of the model's Brewer-Dobson circulation against observations is required in order to justify confidence in the results.

We agree with the reviewer that the vertical resolution of our model in the stratosphere is inadequate to resolve the complex stratospheric dynamics. It is one of the major limitations in our work. Interestingly, Smith et al. (2014) have compared the simulated climate by CAM4 version with a "high-top" WACCM version which has highly resolved stratosphere and mesosphere. They have shown that CAM4 with the limited vertical resolution is able to simulate the Brewer-Dobson circulation, although there are differences when compared with WACCM simulation. This can be seen in Figure 10 of Smith et al., (2014) where the Transformed Eulerian Mean (TEM) vertical winds are shown. We now discuss this limitation of model in the first paragraph of section 3.4.

2. Page 4, lines 26-27. The manuscript at present simply states "The zonal variations as well as interannual variations (for this study) in mixing ratio of the volcanic aerosols are ommitted". Although they do make this clearer later in the Discussion/Conclusion, it needs to be made much clearer here that this means that their model includes no aerosol transport, deposition, microphysics or chemistry - that the aerosol layers are simply represented by fixed, globally-uniform values.

We added this information in section 2.2 of the revised manuscript.

3. Page 7, lines 18-19. What the authors call the "burn-off effect" with reference to Ackerman et al. (2000) is completely irrelevant as an explanation here. Ackerman et al. examined the impact on boundary-layer trade cumuli of low-level soot. This has no bearing on the reduction of upper-tropospheric cirrus cloud being discussed at this point.

4. Page 12, lines 8-9. The authors again use the term "burn-off effect" but this time with reference to Visioni et al. (2018). The term again seems inappropriate as Visioni et al. explain the thinning of high-altitude cirrus clouds in terms of an increase in atmospheric stability and thus a decrease in turbulence and updraft velocities - nothing about "burn- off".

Thank you for pointing this out. We agree and remove the term "burn-off effect". We removed the reference to Ackerman et al. (2000) from the manuscript. We modified the section to include the results from Kuebbeler et al. (2012) and Visioni et al. (2018) in the revised manuscript.

**Technical Corrections/Comments**

1. Page 4, line 13: the number of model layers in the stratosphere should be given.

The number of stratospheric layers is 8. We added this information in the revised manuscript in section 2.1

2. Page 7, line 22-23: the text currently reads "...leads to an increase in low cloud for the Volc_100hPa case relative to the Vol_70hPa and Volc_35hPa cases..." This is not incorrect, but I think it would be clearer to say "...leads to less of a decrease in low cloud for the Volc_100hPa case compared with the Volc_70hPa and Volc_35hPa cases..."

Thanks for the suggestion. We have modified this sentence in the revised manuscript as per the reviewer's suggestion.

3. Page 11, lines 25-26: "autotrophic" is misspelled as "autotropic" three times.

Thank you for pointing out the typo. We have corrected the spelling in the revised manuscript. .

4. Supplementary material, page 6: the caption to Figure S1 should explain what is shown in each of the panels (a) to (f).

We added the information about each panel in the caption of Fig-S1in the revised version.

5. Supplementary material, page 12: the term "1XCO2" is used in the caption to Figure S7 and has been used throughout the paper, but "CTL" is used in the titles of the individual panels; consistency would avoid any confusion.

We have corrected this inconsistency in the figure.

6. Supplementary material, page 13: the values plotted in Figure S8 are presumably global-means?

Yes. The values are global means. We mention this in the caption of the revised version.

References:

Kuebbeler, M., Lohmann, U. and Feichter, J.: Effects of stratospheric sulfate aerosol geo-engineering on cirrus clouds, Geophys. Res. Lett., 39(23), 1–5, doi:10.1029/2012GL053797, 2012.

Smith, K. L., Neely, R. R., Marsh, D. R. and Polvani, L. M.: The Specified Chemistry Whole Atmosphere Community Climate Model (SC-WACCM), J. Adv. Model. Earth Syst., 6(3), 883–901, doi:10.1002/2014MS000346, 2015.

Visioni, D., Pitari, G., di Genova, G., Tilmes, S. and Cionni, I.: Upper tropospheric ice sensitivity to sulfate geoengineering, Atmos. Chem. Phys., 18(20), 14867–14887, doi:10.5194/acp-18-14867-2018, 2018.

---

## Author Response (AR1)

**Reply to the comments from Anonymous Reviewer-1**

I reviewed an earlier version of this manuscript that was submitted to another journal. I think this version is far improved over the previous one. Many of the results are perhaps not all that surprising, especially given that other studies (which the authors cite) have looked at the different effects of injection altitude. However, I have not necessarily seen all of these results in one place, which makes this paper interesting. The addition of Section 3.5 is very interesting, and I learned quite a bit. I am recommending just a few minor revisions.

We thank the reviewer for the time spent on evaluating our manuscript.

**General comments:**

As the authors say, they don't include any dynamics or transport. However, radiative forcing and climate response to stratospheric aerosols definitely depend on dynamics. I would appreciate the authors adding some description as to how this compromise might have affected their results.

We have written that only the transport of aerosols is not modelled. However, the stratospheric dynamics is included in our simulations.

**Specific comments:**

In the last paragraph on Page 5 (going into Page 6), some context for these results is needed. Do these numbers make sense, and why? (I think they make sense, but I'd like for you to say so.)

The TOA radiative imbalance discussed in the paragraph are actually the prescribed-SST radiative forcing as discussed in several previous studies (Bala et al., 2010; Modak et al., 2014; Nalam et al., 2018). This imbalance is corrected for the land surface temperature change in the prescribed SST simulations, to obtain the TOA radiative forcing in the two-point method as discussed in Modak et al., (2018) and Duan et al., (2018) and in the supplemental sect. S1. We discuss this in the revised text.

Page 8, line 13: Can you phrase this in a different way? 1xCO2 is your baseline, so it doesn't cause any cooling.

We have rephrased this line in the revised version as ".....which attains only 70% of the cooling in 1XCO2 relative to 2XCO2."

Figure 4: I'm not sure hatching is necessary. All of the regions are statistically significant, so just say that.

We have adjusted the transparency of the hatching in the revised version.

References:

Bala, G., Caldeira, K. and Nemani, R.: Fast versus slow response in climate change: Implications for the global hydrological cycle, Clim. Dyn., 35(2), 423–434, doi:10.1007/s00382-009-0583-y, 2010.

Duan, L., Cao, L., Bala, G. and Caldeira, K.: Comparison of the Fast and Slow Climate Response to Three Radiation Management Geoengineering Schemes, J. Geophys. Res. Atmos., doi:10.1029/2018JD029034, 2018.

Modak, A. and Bala, G.: Sensitivity of simulated climate to latitudinal distribution of solar insolation reduction in solar radiation management, Atmos. Chem. Phys., 14(15), 7769–7779, doi:10.5194/acp-14-7769-2014, 2014.

Nalam, A., Bala, G. and Modak, A.: Effects of Arctic geoengineering on precipitation in the tropical monsoon regions, Clim. Dyn., 50(9–10), 3375–3395, doi:10.1007/s00382-017-3810-y, 2018.

**Reply to the comments from Anonymous Reviewer-2**

The authors simulate the solar radiation method (SRM) of stratospheric sulfate aerosols by prescribing a uniform layer of sulfate aerosol concentration. Assuming different altitudes of this layer, they determine the impact on different variables like radiative forcing, surface temperature, and humidity and temperature in the stratosphere. The paper is very well written and includes an impressive literature review. I recommend publication after the authors addressed the following comments.

We thank the reviewer for the constructive comments and the time spent on reviewing the manuscript.

**General comments**

My main concern is related to the prescribed aerosol layer. Prescribing aerosols in a climate model is a widely used technique and can be used to be computationally more efficient. However, the prescribed layer should be comparable to reality and preferable been calculated previously with a aerosol microphysical model. This study simplified the method by prescribing the aerosol concentration meridionally uniformly at a certain height (km) above sea level. This distribution is quite unrealistic. However, this may not have extremely strong impact on the main conclusion of the paper, the dependency of radiative forcing on altitude.

Stronger impact has the fact that the prescribed profiles do not include the changes caused by sedimentation. Sedimentation causes a vertical spread of the aerosols, the stronger the higher the injection altitude above the tropopause (e.g. (Tilmes et al. (2017)). This difference to a more realistic profile is not mentioned in the paper. As a consequence, heating of the aerosols will not occur in a small layer as shown in Figure 6a and 7. The heated layer extends to the tropopause. This has consequences for clouds and humidity transport. This aspect is not at all discussed in the paper.

I agree with the authors that a simplification is useful and can help to gain new insight. The dependency of the radiative forcing on the altitude of the SRM agent has been shown for three different SRM techniques by Niemeier et al (2013). Sulfur injection studies have shown that the resulting radiative forcing depends on the altitude of the injection. This has mostly been related to the vertical extension of the aerosol layer. This paper could give new insights by adding a simulation with more realistic sulfate profile. This would show whether the altitude of the injection or the vertical extension of the aerosol layer is the cause of an increased radiative forcing.

The limitation of the model, no aerosol microphysics, are well described. The more important limitation of the prescribed sulfate concentration are not prescribed. Please, add a figure of the prescribed sulfate layers in the main article. For a reader who is not familiar with stratospheric sulfate distribution it would be very helpful.

Thanks for these comments. We agree that sedimentation and a realistic distribution of aerosols in the vertical and related radiative heating distribution is important and they are missing in our study. However, we believe, as the reviewer has also pointed out, that the qualitative and fundamental effects related to the height of the aerosols will not be altered. In this paper, our main aim is to investigate sensitivity to the height of aerosol layer and not the height of aerosol injection which brings additional complexity by spreading the aerosols in

the vertical. We plan to use realistic vertical distributions corresponding to various heights of injections in a future study. These limitations are discussed in a paragraph in the last section in the revised manuscript. The height sensitivity studied by Niemeier et al (2013) is discussed in the introduction. The figure as suggested by the reviewer is also included in the revised manuscript (Figure 1 in the revised manuscript).

**Specific comments**

Page 3 Line 27: I don't understand this sentence.

We rewrote the sentence in the revised version to remove the ambiguity. It is revised as "The altitude of the prescribed aerosol layer can thus affect climate depending on the proximity of the heated layer to the tropopause as heat exchange between stratosphere and troposphere can lead to changes in clouds and stratospheric water vapor"

Section 2.2: How is the sulfate layer created? The global distribution in Nalam et al (2017) is unrealistic. Aerosols follow tropopause in a meridional cross section (e.g. Visioni et al. (2017) or Dhomse et al. (2014)). Add figures of sulfate distribution. It remains unclear how the sulfate is distributed vertically. I assume you only change the height of the prescribed layer.

Yes, the aerosols are added in single layers as discussed at the end of the first paragraph in section 2.2. See our response to the general comments. A figure showing the aerosol distribution is also included in the revised manuscript.

Page 4 Line 22: 20 Mt SO4? Change unit to Tg (SI unit)

We have modified the units throughout the manuscript to Tg.

Page 6 Line16: Thus, CO2 reduction would be most effective. This would be a good statement in the conclusion.

Thank you for the suggestion. Yes, the efficacy of sulfates is less than one and for equivalent change in radiative forcing  $CO_2$  reduction will be more effective. We discuss this in the revised manuscript.

Page 7 Line 16 - 26: Results of Kuebbeler et al (2012) should be taken into account.

The results from Kuebbeler et al (2012) are now discussed in the revised manuscript in section 3.2.

Page 8 Line 30: See also Liepert and Prevedi (2009) and Kravitz et al (2013).

Thank you for the references. We have cited these references in the revised version.

Page 8 Line 27pp: This is a very short summary of this topic.

We have added more discussion on fast adjustments and related precipitation in this section in the revised manuscript.

Page 9 Stratospheric dynamics: The vertical resolution of you model is very limited. Good results of stratospheric dynamics need a higher resolution, even in a model that is still not capable to generate a QBO. You should mention the QBO in the first paragraph and not only cite the two papers.

We have added more discussion in the first paragraph of section 3.4 in the revised version to discuss this limitation of our model.

Page 9 Line 27: You show significance for the temperature. How about the zonal wind? Your differences are very small. Please add a figure with significance of the zonal wind.

As suggested by the reviewer, a sperate figure of zonal wind is added in the revised manuscript (Fig-7 in revised manuscript). The statistically significant wind changes shown as hatched areas in the figure.

Page 9 Line 29: Higher temperature in the tropics because of more solar radiation. The aerosols cannot absorb as much radiation during polar winter then during summer.

We have modified the line to include this information.

Page 10 Line 3: Significant?

Yes. Significant at 5% significance level, as shown in the revised figure 7.

Page 10 Line 4 to 10: Your vertical profile of aerosols is unrealistic. As a consequence the vertical profile of temperature anomaly and humidity anomaly are unrealistic. You have to say this and the consequences for your results.

We have discussed the consequence of the unrealistic aerosol profile on temperature and humidity distribution in a paragraph in the last section in the revised manuscript.

Page 12 Line 5 to 6: I don't understand this sentence? What would be the relation between microphysics and global mean surface temperature?

The aerosol microphysical changes can affect the optical and radiative properties of the aerosols through nucleation, condensation, coagulation, hygroscopic growth, etc (e.g. Heckendron et al., 2009). The changes in the optical and radiative properties affects the radiative forcing and thereby influences the surface cooling efficiency in the aerosol geoengineering schemes. This is discussed in the revised manuscript.

Page 12 Line 15-16: This is not a new result and can be found in many previous studies. I don't think there is a tendency to include radiation to aerosol coupling for SW only in the models, at least not in general.

We agree with the reviewer that this result is not new. We delete the phrase "and the need for carefully accounting LW forcing along with the SW forcing"

Page 12 Line 25 Many studies show that higher altitude of injection causes stronger forcing. Most studies relate this to the thicker aerosol layer. Niemeier et al. (2013) have show that for different SRM methods the forcing depends on the altitude of the forcing agent. Thus, your result is not new. It remains open, whether the top level of the aerosols or the vertical extension of the sulfate layer is more important. Your study would gain a lot if you could answer this by adding a simulation with a more realistic vertical profile.

We appreciate the comment. We have discussed the results from Niemeier et al. (2013) in the introduction of the revised manuscript and the discussed the limitation due to vertical aerosol extension in the last section. Previous studies have estimated the climate sensitivity to altitude mainly based on the sedimentation effects and transportation. In our approach, the primary focus was to estimate the changes in effective radiative forcing due to changes in fast adjustment processes (e.g., Boucher et al., 2017) when the aerosol induced warm layer moves away or closer to the tropopause. As mentioned in response to the general comments, we plan to study the effects of realistic profile in a future study.

Page 13 Line 15 to 25: As stated above, the main limitation is the profile itself. The profile changes when injecting at higher altitude and the particle sediment. This has to be reflected in your profile and would change e.g. vertical humidity transport.

We agree. We have discussed this limitation and its consequence related to the idealized profile in a paragraph in the revised "Discussion and conclusion" section.


Thank you for pointing out the typo. We have corrected the spelling in the revised manuscript. .

4. Supplementary material, page 6: the caption to Figure S1 should explain what is shown in each of the panels (a) to (f).

We added the information about each panel in the caption of Fig-S1in the revised version.

5. Supplementary material, page 12: the term "1XCO2" is used in the caption to Figure S7 and has been used throughout the paper, but "CTL" is used in the titles of the individual panels; consistency would avoid any confusion.

We have corrected this inconsistency in the figure.

6. Supplementary material, page 13: the values plotted in Figure S8 are presumably global-means?

Yes. The values are global means. We mention this in the caption of the revised version.

The changes in the revised manuscript and supplementary files are shown in the attached marked-up version.

[revised manuscript text omitted]

---

## Referee Report (RR1)

Review on:
**Climate System Response to Stratospheric Sulfate Aerosols: Sensitivity to Altitude of Aerosol Layer**
Krishna-Pillai Sukumara-Pillai et al

The authors simulate the solar radiation method (SRM) of stratospheric sulfate aerosols by prescribing a uniform layer of sulfate aerosol concentration. Assuming different altitudes of this layer, they determine the impact on different variables like radiative forcing, surface temperature, and humidity and temperature in the stratosphere.

The paper is very well written and includes an impressive literature review. I recommend publication after the authors addressed the following comments.

**General comments**

I expressed my concern related to the prescribed aerosol layer in my first review. The authors have not performed a more realistic simulation, where the sulfate spreads over a larger vertical area. They want to keep this for another more detailed study. As they state this simplification know clearly in the text, I can accept this. However, I am not really satisfied with the motivation of the work. The results of Kleinschmitt et al (2017) are different to previous studies, as one can see also in the comparison to the second model in their paper. Tilmes et al (2017), English et al, Niemeier and Schmidt (2017) show an increase of the TOA imbalance when increasing the injection height. Thus, motivating the work with Kleinschmitt et al (2018) is difficult in my point of view.

**Specific comments**

Check the spelling of names in citations.

Page 3 Line 9 Why depends sedimentation on Brewer-Dobson Circulation (BDC)? Sedimentation is vertical and BDC mostly meridional transport, at least the deep branch as named in the text.

Page 10 Line 23 - 26: This needs to be sorted! Aquilla shows this for tropical jets of the QBO. You model has no QBO and strengthens the easterly jets in this area. You seem to mix tropical and high latitude jets here. Additionally, the largest negative response in in Vol_100hPa.

Fig. 8: Include the upper troposphere region in the plot.

Page 13 Line 10: This is not new and shown in many more previous studies.

results depend on the model, strong uncertainty, but also on injection strategy and model resolution. Resolving the QBO or not has an impact as well.

Page 14 Line 3-4: What about the easterlies. You never say that your westerlies are the polar night jet, not the tropical jets. A model without QBO gives easterlies in the tropics.

---

## Author Response (AR2)

**Reply to the comments from Anonymous Reviewer**

The authors simulate the solar radiation method (SRM) of stratospheric sulfate aerosols by prescribing a uniform layer of sulfate aerosol concentration. Assuming different altitudes of this layer, they determine the impact on different variables like radiative forcing, surface temperature, and humidity and temperature in the stratosphere.

The paper is very well written and includes an impressive literature review. I recommend publication after the authors addressed the following comments.

We thank the reviewer for the suggestions which helped us to further improve the manuscript.

**General comments:**

I expressed my concern related to the prescribed aerosol layer in my first review. The authors have not performed a more realistic simulation, where the sulfate spreads over a larger vertical area. They want to keep this for another more detailed study. As they state this simplification know clearly in the text, I can accept this. However, I am not really satisfied with the motivation of the work. The results of Kleinschmitt et al (2017) are different to previous studies, as one can see also in the comparison to the second model in their paper. Tilmes et al (2017), English et al, Niemeier and Schmidt (2017) show an increase of the TOA imbalance when increasing the injection height. Thus, motivating the work with Kleinschmitt et al (2018) is difficult in my point of view.

Models used in the several previous simulate complex processes such as aerosol microphysics, chemistry, sedimentation, transport, etc. Thus, the altitude sensitivity estimated in these studies is a net effect of all these processes. The estimation of individual contribution to the sensitivity is lacking. In our approach, we used an idealized prescribed aerosol model which has the advantage of isolating and analyzing the individual effects of these processes. In this study, by fixing the aerosol size and amount, we analyzed the altitude sensitivity of climate related to the fast adjustment processes such as aerosol-induced heating and the consequent changes in stratospheric water vapor and clouds (Boucher et al., 2017). We have modified the last paragraph of the introduction section in the revised manuscript to clearly communicate this motivation.

**Specific comments:**

Check the spelling of names in citations.

Thank you for pointing out the typos. We have corrected the errors in the revised version.

Page 3 Line 9 Why depends sedimentation on Brewer-Dobson Circulation (BDC)? Sedimentation is vertical and BDC mostly meridional transport, at least the deep branch as named in the text.

For the high altitude injection at the equator, Tilmes et al., (2017) shows that the upward branch of the BDC in the tropics carries the aerosols upwards which results in maximum stratospheric sulfate burden in the tropics for all seasons. For the injections outside the tropics, the meridional transport is significant, where the aerosol follows the deep branch of

BDC and is transported more effectively toward middle and high latitudes. We have modified this section in the revised manuscript to include these details.

Page 10 Line 23 - 26: This needs to be sorted! Aquilla shows this for tropical jets of the QBO. You model has no QBO and strengthens the easterly jets in this area. You seem to mix tropical and high latitude jets here. Additionally, the largest negative response in in Vol_100hPa.

Thank you for the suggestion. We have modified this paragraph in the revised manuscript to discuss the changes in tropical and high latitude jets separately.

Fig. 8: Include the upper troposphere region in the plot.

The figure is modified in the revised manuscript to include the upper troposphere region.

Page 13 Line 10: This is not new and shown in many more previous studies. results depend on the model, strong uncertainty, but also on injection strategy and model resolution. Resolving the QBO or not has an impact as well.

We have modified the sentence in the revised manuscript to include more details as suggested.

Page 14 Line 3-4: What about the easterlies. You never say that your westerlies are the polar night jet, not the tropical jets. A model without QBO gives easterlies in the tropics.

The sentence is modified in the revised manuscript to "The aerosol induced stratospheric warming and the resulting strong stratospheric high-latitude westerly and tropical easterly wind anomalies are sensitive to the altitude of the aerosols".

**List of relevant changes in the revised manuscript:**

Dear Editor and Reviewers,

The manuscript have been corrected by the following modifications.

1) Throughout the manuscript, modifications are made as per the suggestions of the reviewer.

2) As suggested by the reviewer, vertical profiles for changes in temperature and specific humidity is modified to include the upper troposphere (Figure. 8 in the revised manuscript).

The changes in the revised manuscript are shown in the attached marked-up version.

[revised manuscript text omitted]